# Progressive Ocular Axial Elongation and Retinal Ganglion Cell Degeneration in Mice with Elastic Fiber Disorder

**DOI:** 10.3390/ijms26189221

**Published:** 2025-09-21

**Authors:** Samuel Insignares, John Kuchtey, Rachel W. Kuchtey

**Affiliations:** 1Vanderbilt Eye Institute, Vanderbilt University Medical Center, 2311 Pierce Avenue, Nashville, TN 37232, USA; samuel.insignares@vumc.org (S.I.); john.kuchtey@vumc.org (J.K.); 2Department of Molecular Physiology and Biophysics, Vanderbilt University, Nashville, TN 37232, USA

**Keywords:** myopia, glaucoma, elastic fiber disorder

## Abstract

We previously reported ocular phenotypes of 1-year-old 129S1/SvlmJ lysyl oxidase-like 1 null (*Loxl1*^−/−^) mice. Here we sought to characterize age-dependent changes in C57BL/6J *Loxl1*^−/−^ mice in a longitudinal fashion. Retinal ganglion cell (RGC) function was assessed by electroretinography (ERG), and optic nerves were evaluated by histological analysis. Ocular biometric measurements were obtained by optical coherence tomography (OCT). We detected reduced RGC function, revealed by decreased amplitude and increased latency of ERG positive scotopic threshold responses (pSTRs) in *Loxl1*^−/−^ mice compared to age-matched wt mice. In addition, there is significant inter-eye asymmetry of RGC function, as well as age-related RGC function loss observed only in *Loxl1*^−/−^ mice. Histologically, we observed enlarged optic nerve areas in *Loxl1*^−/−^ mice compared to wt mice. Significant ocular biometric differences between two groups were detected, most notably, age-related axial elongation of the globe, accompanied by deepening of anterior chamber depth (ACD). Though eyes elongate with age in both groups, this is more pronounced in *Loxl1*^−/−^ mice, and the elongation of the globe correlated with decreased RGC function. The correlation of age-related reduction in RGC function with globe axial elongation may have implications for the association of axial myopia with glaucoma and aging in humans.

## 1. Introduction

Glaucoma is a leading cause of irreversible blindness due to progressive degeneration of RGCs and their axons. While the final pathway is apoptosis of RGCs, the initial site of damage occurs at the optic nerve head (OHN) [1]. Deformation of the ONH may lead to axon transport deficits, RGC functional loss, and eventual cell death [2]. Many risk factors for glaucoma have been identified, and the association between myopia and glaucoma has been increasingly recognized [3]. Although the precise mechanism for such an association remains unknown, a plausible hypothesis linking these two conditions is aberrant connective tissue remodeling of the ocular shell, resulting in globe elongation in axial myopia and ONH biomechanical changes in glaucoma [4].

One particular form of glaucoma, exfoliation glaucoma (XFG), is caused by exfoliation syndrome (XFS), which is a systemic connective tissue disorder affecting many organs including the eyes [5]. A landmark genome-wide association study identified genomic variants of *LOXL1* associated with XFS/XFG [6]. LOXL1 is a member of the lysyl oxidase family, which is composed of lysyl oxidase (LOX) and lysyl oxidase-like 1-4 (LOXL1-4), all of which play important roles in the maintenance of extracellular matrix-rich tissue [7]. Specifically, LOXL1 is essential for crosslink formation in elastin and collagen [8]. Previous studies using LOXL1-knockout mice (*Loxl1*^−/−^) revealed characteristic findings of elastic fiber disorder, such as pelvic organ prolapse (POP) and laxity of the skin, which established *Loxl1*^−/−^ mice as a rodent model for investigating elastic fiber maintenance and homeostasis [9]. We recently investigated ocular phenotypes of *Loxl1*^−/−^ mice on the 129S1/SvImJ background at 1 year of age using multimodal technologies, including spectral domain optical coherence tomography (SD-OCT), transmission electron microscopy (TEM), and atomic force microscopy (AFM) [10]. We discovered that these mice had biometric changes in the anterior segment of the eye and enlarged optic nerves with stiffer posterior sclera that correlated with enlarged collagen fibril diameter, reduced elastin deposition, and irregular elastic fiber morphology. In this current study, we report characterization of ocular features of our newly backcrossed *Loxl1*^−/−^ mice on the C57BL/6J genetic background. Our findings with 1-year-old *Loxl1*^−/−^ mice on the C57BL/6J background were consistent with our previous report [10]. In addition, we followed *Loxl1*^−/−^ mice on the C57BL/6J background longitudinally from 4 to 16 months of age. We discovered progressive globe elongation correlating with RGC degeneration compared with age-matched wt mice, providing further evidence of the association between axial myopia and glaucoma.

## 2. Results

### 2.1. Loxl1^−/−^ Mice Have Expanded Optic Nerves

In experimental rodent glaucoma models with elevated intraocular pressure (IOP), optic nerve expansion occurs earlier than loss of retinal ganglion cells (RGCs) [11,12]. We have previously reported this phenotype in microfibril-deficient mice without elevated IOP [13,14] and in 1-year-old *Loxl1*^−/−^ mice on the 129S1/SvImJ background [10]. Therefore, in this study, we evaluated the optic nerves of C57BL/6J *Loxl1*^−/−^ mice using our established protocols. As shown in Figure 1A, consistent with our previous findings with 129S1/SvImJ mice, 1-year-old *Loxl1*^−/−^ mice on the C57BL/6J background had expanded optic nerves compared with wt mice. Additionally, this phenomenon was seen in C57BL/6J mice as young as 4 months old. To determine if the expanded optic nerve precedes loss of axons, as shown in other glaucoma models with elevated IOP, we counted axons using two methodologies, manual counting (Figure 1B) and automated counting with AxoNet2.0. By manual counting, we did not detect loss of axons in *Loxl1*^−/−^ mice at either 4 or 12 months of age when compared with wt mice (Figure 1B), although axon density trended towards being significantly reduced in *Loxl1*^−/−^ mice (Figure 1C) at 12 months old, likely driven by the larger nerve areas. Similar results were obtained by automated counting with Axonet 2.0. These findings indicate that reduced axon density is not due to RGC axon loss but is rather due to optic nerve expansion.

### 2.2. Loxl1^−/−^ Mice Have Reduced RGC Function

RGC dysfunction in glaucoma has been shown to be reversible by aggressive lowering of IOP, resulting in improved visual field parameters in human glaucoma patients [15]. This provided the premise for us to examine RGC function in *Loxl1*^−/−^ mice, even though there was no detectable loss of axons at 1 year of age. We performed scotopic ERGs in *Loxl1*^−/−^ and wt mice (Figure 2A–C). ERG waveforms were stimulated by flashes at a low stimulus intensity of −5 log cd·s/m^2^ to elicit the positive scotopic threshold response (pSTR), which is a representation of RGC function [13,16]. *Loxl1*^−/−^ mice have a significantly reduced pSTR amplitude (Figure 2D) as well as a significant increase in pSTR latency compared to wt mice at 12 months of age (Figure 2F), suggesting RGC functional deficit preceding RGC axon loss.

In addition, asymmetry between the two eyes in each individual glaucoma patient is a known phenomenon, especially in early stages of the disease [17]. We observed significant differences between the *Loxl1*^−/−^ and wt groups, as well as a large variation in both pSTR amplitude and latency in *Loxl1*^−/−^ mice (Figure 2D,F). Therefore, we analyzed asymmetries by comparing the better performing eye to the worse performing eye in both *Loxl1*^−/−^ and wt mice. In Figure 2E, the better eye is defined as the eye with the higher pSTR amplitude, and the better eye in Figure 2G is defined as the eye with shorter latency from the stimulus. Interestingly, we detected significant differences between the better eye and worse eye for both pSTR amplitude and latency (Figure 2E,G), which were not observed in wt mice, consistent with glaucomatous RGC damage.

### 2.3. Loxl1^−/−^ Mice Have Accelerated Age-Related Decline of RGC Function

Glaucoma is an age-related neurodegenerative disease characterized by loss of RGCs [18]. We next examined if the reduced RGC function in *Loxl1*^−/−^ mice was age-related. We aged our C57BL/6J cohort to 16 months and measured pSTR in *Loxl1*^−/−^ and wt mice at 4, 12, and 16 months of age. *Loxl1*^−/−^ showed significantly greater age-related reduction in pSTR amplitude (*p* = 0.001) and prolonged latency (*p* = 0.007) compared with wt mice (Figure 3), indicating the acceleration of age-related decline of RGC function.

### 2.4. Loxl1^−/−^ Mice Have Significant Anterior Segment Biometric Changes

Previously, with 1-year-old mice on the 129S1/SvImJ background, we found that central cornea thickness (CCT) and ACD were significantly different in *Loxl1*^−/−^ compared to wt mice, although no change was noted in AL [10]. Subsequently, as shown in our separate publication, with 4-month-old mice on the C57BL/6J background, all three parameters (CCT, ACD, and AL) were significantly different when comparing *Loxl1*^−/−^ to wt mice [19]. In this current study, we increased the number of 4-month-old mice and confirmed the previously published findings. More importantly, we followed those mice longitudinally to 12 months and 16 months of age. We observed significant differences in Loxl1^−/−^ compared to wt mice in CCT and ACD, except in CCT at 16 months (Figure 4). We did not observe body weight differences between *Loxl1*^−/−^ and wt mice. It is interesting that CCT thinning was observed as early as 4 months of age in *Loxl1*^−/−^ compared to wt mice and maintained statistical difference to 12 months of age. No age-related change in CCT was observed in wt mice throughout (Figure 4A); however, an increase in CCT at 16 months compared with 4 and 12 months of age was seen in *Loxl1*^−/−^ mice. *Loxl1*^−/−^ mice also had deeper ACD compared with wt mice (Figure 4B), consistent with the axial myopia phenotype in human patients. The *Loxl1*^−/−^ ACD also showed statistically significant age-related deepening compared to wt mice (Figure 4B’). Both *Loxl1*^−/−^ and wt mice showed age-related increases in lens thickness (Figure 4C), resembling human patients, but there was no difference between groups at any age (Figure 4C’).

It is well-known that ocular biometric measurements differ between males and females in humans [20]. We therefore analyzed all OCT measurements, including CCT, ACD, LT, and AL in male and female mice at 4 months old separately, given the sufficiently large number of mice at this age in each group. Interestingly, we detected thinner CCT in *Loxl1*^−/−^ mice, but thicker LT and longer AL in wt mice. The other measurements showed no differences related to sex (Figure 5A’,C’,D’).

### 2.5. Loxl1^−/−^ Mice Have Globe Elongation, Which Is Correlated with Decreased RGC Function

In addition to the anterior segment biometry findings presented above, we also observed statistically significant globe elongation, shown as increased AL in *Loxl1*^−/−^ compared to wt mice at all three ages (Figure 4D). Age-related globe elongation was also shown, with this effect more pronounced in *Loxl1*^−/−^ than in wt mice (Figure 4D’), demonstrated as a statistically steeper slope of the linear regression (*p* = 0.005). Consistent with biometric findings in myopic human eyes [21], the significant age-related increase in AL in *Loxl1*^−/−^ mice was accompanied by significantly deeper age-related ACD (Figure 4B’,D’).

The association between axial myopia and glaucoma has been well demonstrated in human patients. To test this hypothesis in a rodent model, we next investigated the correlation between globe elongation and RGC function (Figure 6). Overall, analysis of combined 12- and 16-month-data from both genotypes showed that increased AL was correlated with decreased pSTR amplitude (*p* = 0.0007). Latency was not correlated with globe elongation (*p* = 0.15).

### 2.6. No Changes in IOP or Visual Acuity (VA) in Loxl1^−/−^Mice

IOP, measured by a rebound tonometer at 4, 12, and 16 months of age, was not different between the *Loxl1*^−/−^ and wt mice, and over time IOP remained constant (Figure 7A). Visual acuity was assessed for the same age groups using the OptoDrum. There were no differences between genotypes in VA at any age, nor were there age-related changes (Figure 7B).

## 3. Discussion

The first *Loxl1*-knockout mice were created in a landmark study by Liu et al. over 2 decades ago, in which the investigators provided strong evidence of elastic fiber defects in the urogenital tract, lungs, and skin of *Loxl1*^−/−^ mice [9]. We obtained these *Loxl1*^−/−^ mice from Dr. Tiansen Li and investigated their genetic background by genome scan SNP analysis in collaboration with The Jackson Laboratory. The results of the scan were consistent with the *Loxl1*^−/−^ mice being on the 129S1/SvImJ background. Our initial study characterized their ocular phenotypes and ultrastructural and biomechanical features in comparison with background-matched wt controls [10]. Wiggs et al. also reported ocular phenotypes of *Loxl1*^−/−^ mice from the same source as we obtained ours, but their comparison was made with C57BL wt mice [22]. Alteration of phenotypes induced by genetic background in mice is common, and this phenomenon was particularly reported in *Loxl1*^−/−^ mice on C57BL/6 vs. 129S genetic backgrounds [23]. To avoid this confounding factor, we backcrossed 129S1/SvlmJ *Loxl1*^−/−^ mice to C57BL/6J mice for 10 generations. All comparisons in the present study were made with age- and background-matched wt controls. In addition, to our knowledge, this is the first longitudinal follow up of *Loxl1*^−/−^ mice at 4, 12, and 16 months of age.

In the current study, we detected reduced retina function, as evidenced by ERG experiments showing decreased amplitudes and increased latencies of the pSTR, which is generated by RGCs [16,24], in *Loxl1*^−/−^ mice at 12 months of age (Figure 2). To fully take advantage of the longitudinal nature of this study, we analyzed the trend of pSTR over time in both *Loxl1*^−/−^ and wt mice (Figure 3). *Loxl1*^−/−^ mice showed progressive decline of RGC function, demonstrated by age-related reductions in pSTR amplitude and prolonged latency, and both were significant in comparison to wt mice. It is noteworthy that glaucoma is an age-related disease, and the hallmark is loss of RGC function leading to RGC soma apoptosis and eventually irreversible loss of vision [25]. Our longitudinal findings of age-related reductions in RGC function in *Loxl1*^−/−^ mice recapitulates the human glaucoma phenotype. Furthermore, asymmetry of optic nerve damage in glaucoma is a well-recognized clinical feature [26]. Sullivan-Mee et al., in a prospective, cross-sectional cohort study, reported diagnostic precision of retina structural asymmetry parameters for identifying early primary open-angle glaucoma [27]. In the Ocular Hypertension Treatment Study (OHTS), the investigators found that, when threshold asymmetry of visual field between eyes existed, the eye with lower thresholds had a significantly greater risk of the development of glaucoma [17]. To investigate the inter-eye difference in RGC function, we compared better eyes vs. worse eyes and detected significant asymmetry for both pSTR amplitude and latency in *Loxl1*^−/−^ mice, but not in wt mice, further supporting a glaucomatous phenotype of *Loxl1*^−/−^ mice.

We previously reported that 129S1/SvImJ *Loxl1*^−/−^ mice had enlarged optic nerves at 12 months old, an early sign of glaucoma in rodents [10]. Consistently, we also observed enlarged optic nerve at both 4 and 12 months of age in C57BL/6J *Loxl1*^−/−^ mice. Although the total axon numbers of *Loxl1*^−/−^ and wt mice were similar at 12 months of age, the axon density was borderline reduced in *Loxl1*^−/−^ mice, likely due to their enlarged optic nerves, all of which indicate a mild glaucoma phenotype. It is worth noting that optic disk size in relation to glaucoma development has been controversial, as Jonas et al. showed glaucomatous optic nerve fiber loss independent of optic disk size [28], whereas another study by the same group of investigators showed that smaller disk size is a risk factor for glaucoma development [29], likely due to differences in patient population. Using a subset of patients with ocular hypertension from OHTS participants, Weinreb et al. previously demonstrated that larger optic disk size, measured by OCT, was significantly associated with the development of glaucoma [30]. One recent study by Nishida et al. revealed that larger optic disk size is associated with faster retinal nerve fiber thinning in glaucoma [31]. The findings of larger optic nerve size in *Loxl1*^−/−^ mice, especially that their enlarged optic nerves precede detectable RGC functional loss, not only supports the notion that larger optic nerve size is an early sign of glaucomatous nerve damage, but it may also suggest mechanistic causality. It is import to note that the most significant risk factor for glaucoma is elevated IOP [25]. The mild nature of this glaucoma phenotype in *Loxl1*^−/−^ mice is not surprising given the fact that the *Loxl1*^−/−^ mice did not have elevated IOP at any age. It is conceivable that *Loxl1*^−/−^ mice will present worsening glaucoma phenotypes if their IOP is experimentally elevated, which is a subject for future investigation. If proven true, it will lend further support to the notion proposed by Burgoyne et al. that a higher pressure gradient across the lamina cribrosa in the ONH in larger optic disks exerts more stress, leading to axonal damage in glaucoma [1].

The predominant role of LOXL1 and other LOX family members is to regulate elastin and collagen crosslinking [8]. LOXL1-deficient mice have loose skin and POP due to elastic fiber defects [9]. We previously showed that *Loxl1*^−/−^ mice on the 129S1/SvImJ background had thin central corneas, deep anterior chamber depths, and enlarged optic nerves that correlated with reduced elastin and morphological changes in collagen fibrils in the peripapillary sclera [10]. Several connective tissue diseases, such as Marfan syndrome and osteogenesis imperfecta, result in thin corneas [32,33]. The finding of thin corneas in *Loxl1*^−/−^ mice revealed in this study is in line with ocular findings of connective tissue disease overall. Moreover, our OCT data also revealed deepening of the ACD in *Loxl1*^−/−^ mice at all three ages. Although age-related deepening is seen in wt mice, it is more pronounced in *Loxl1*^−/−^ mice (Figure 4B’). We also observed age-related thickening of the lens in both *Loxl1*^−/−^ and wt mice, but no difference between groups at any age was seen. This indicates that *Loxl1*^−/−^ mice did not have microspherophakia, a phenotype seen in other connective tissue diseases [34]. We offer two possible explanations for the significantly increased ACD in *Loxl1*^−/−^ mice compared with wt mice. First, it could be the result of posterior movement of the lens due to compromised lens zonular fibers. This phenotype resembles ectopia lentis, a hallmark diagnostic feature in Marfan syndrome [35]. The most prominent component of the lens zonular fibers are fibrillins, but a proteomic study in human and bovine lens zonules also identified LOXL1 [36]. The lack of LOXL1 in our model could account for compromised zonular fibers that manifest as a deeper ACD. One of the hallmarks of XFS is the dislocation of the lens due to compromised zonular fibers [37], providing evidence of *Loxl1*^−/−^ mice as an appropriate model to understand connective tissue diseases, including ocular manifestation of XFS.

The second explanation is related to the increased AL in *Loxl1*^−/−^ mice. The most significant finding of *Loxl1*^−/−^ mice in this study is progressive globe elongation. Consistent with a previous report, globe size increases with age, as shown in wt C57BL/6J mice [38]. However, it is notable that the *Loxl1*^−/−^ mice had longer ALs than wt mice at all three ages, and the increased ACD in *Loxl1*^−/−^ mice is related to increased AL, as commonly seen in myopic human eyes [21]. More significantly, the age-related increase in AL in *Loxl1*^−/−^ mice was statistically significant compared to wt mice, as shown in Figure 4D,D’. The progressive globe elongation correlates with a progressive reduction in RGC function (Figure 5). The correlation of axial myopia and glaucoma has been extensively investigated, and myopia has been well recognized as a significant risk factor for glaucoma development and progression [3,39,40,41], including in a recent dose–response meta-analysis [42]. While we could not conclude the causality of the lack of LOXL1 in either axial myopia or glaucoma, the correlation of axial elongation and reduced RGC function further supports the notion of *Loxl1*^−/−^ mice as an excellent rodent model to understand the association between glaucoma and myopia.

## 4. Materials and Methods

### 4.1. Animals

All animal experiments were performed in accordance with the Association for Research in Vision and Ophthalmology statement for the Use of Animals in Ophthalmic and Vision Research and with approval by the Institutional Animal Care and Use Committee of Vanderbilt University (M1800053-02; 14 March 2024). *Loxl1*^−/−^ mice obtained from Dr. Tiansen Li [9], confirmed by genome scan SNP analysis (The Jackson Laboratory, Bar Harbor, ME, USA) to be on the 129S1/SvImJ background, were backcrossed to the C57BL/6J background over 10 generations. Animals were housed in a facility operated by the Vanderbilt University Division of Animal Care, with a 12 h light cycle and ad libidum access to standard mouse chow and water. The light cycle was kept consistent through their life span, except during the overnight dark adaption period prior to electroretinogram experiments. *Loxl1^+/−^* mice were bred to produce cohorts of *Loxl1*^−/−^ experimental animals and wild-type (wt, *Loxl1^+/+^*) contemporaneous littermates which served as controls. As previously described [9,19], a genotyping service was provided by Transnetyx (Memphis, TN, USA) to identify *Loxl1^+/+^* and *Loxl1*^−/−^ mice, which was validated by PCR reaction on ear-clip DNA using primers S32 (5′-ACA CGT CGG TGC TGG GAT CA-3′), D5 (5′-CTT TCG TAA ACC AGT ATG AGA ACT ACG ATC-3′), and N5 (5′-CGA GAT CAG CCT CTG TTC CAC-3′) (IDT, Coralville, IA, USA).

A total of 125 mouse eyes were included in the experiments, and the number of sub-sets of eyes for each experiment in each genotype and age group are indicated in Figure 1, Figure 2, Figure 4 and Figure 6.

### 4.2. Intraocular Pressure (IOP) Measurements

Mice were anesthetized with 2.5% isoflurane in 95% O_2_/5% CO_2_ delivered at 1.5 L/min by an inhalation anesthesia system (Vet Equip, Livermore, CA, USA). IOP was measured using a TonoLab rebound tonometer (iCare, Franconia, NH, USA), with measurements completed within two minutes of loss of consciousness induced by isoflurane to avoid anesthetic effects [43] and at the same time of day (between 3 p.m. and 4 p.m.) to avoid diurnal fluctuations [44]. Three average readings, each consisting of the mean of six consecutive error-free readings, were averaged to obtain a single IOP value per eye. One eye was reported per mouse.

### 4.3. Visual Acuity via Optomotor Reflex

Photopic visual acuity of mice was assessed using the OptoDrum (Stria.Tech, Tübingen, Germany). Mice were placed, unrestrained, on an elevated platform centered among four adjoining LCD screens on which alternating black and white vertical stripes at 100% contrast traveled in either a clockwise or counterclockwise direction at temporal frequency of 12°/s. The spatial frequency (cycles/degree) of the presented stimulus was altered by adjustment of stripe widths. Visual acuity was determined as the lowest spatial frequency that elicited the optomotor reflex (head tracking). Mice were acclimated in the testing arena for five minutes before initiation of testing. Tests were performed once for each mouse, and results from each eye were considered independently.

### 4.4. Spectral Domain Ocular Coherence Tomography (SD-OCT)

SD-OCT was carried out as previously described [45]. Mice were anesthetized with ketamine/xylazine (100/7 mg/kg). One eye from each mouse was visualized using the Bioptigen Envisu R2200 SD-OCT system using a “mouse retina” lens (Leica Microsystems, Wetzlar, Germany). The pupil was dilated with 1% tropicamide and the mouse was placed in the mouse holder. Genteal Tears Lubricant Eye Gel (Alcon, Fort Worth, TX, USA), wiped across the whiskers and eye, was applied to keep whiskers and eye lashes away from the light path and to keep the eye moist. Mouse position was adjusted until Purkinje lines were perpendicular and parallel to the visual axis and centered on the corneal surface. Axial length (AL) was determined by the acquisition of a series of three central B-scan images to determine (1) the distance from the outer retinal pigment epithelium to the posterior surface of the lens [vitreous cavity (VC) + retina]; (2) the distance from the outer corneal surface to the anterior surface of the lens (central cornea thickness [CCT] + anterior chamber depth [ACD]); (3) half of the lens axial thickness (half lens), determined by an image in which the lens was optically folded in half. AL was defined as equal to (VC + retina) + (CCT+ ACD) + 2 (half lens). Measurements were determined using the digital caliper function of the InVivoVue Software version LAS X 5.3.1 (Leica Microsystems). Image acquisition was completed before lens opacity or corneal damage appeared [46]. Upon completion of imaging, mice were injected with atipamezole (1 mg/kg; Patterson Veterinary, Greeley, CO, USA) to reverse anesthesia and to prevent xylazine-induced corneal damage [47].

### 4.5. Scotopic Flash Electroretinograms (ERGs)

Scotopic flash ERGs were performed using the Celeris system (Diagnosis, Lowell, MA, USA). Mice were dark-adapted overnight and prepared for recordings under dim red illumination. Mice were anesthetized with ketamine/xylazine/urethane (28/11.2/560 mg/kg), placed on the Celeris system, and had a ground electrode inserted subcutaneously at the flank. Pupils were dilated with 1% tropicamide for 5 min. Genteal Tears Lubricant Eye Gel (Alcon) was applied in the same fashion as for OCT. Dim stimulator electrodes were placed in contact with the corneas. Mice were stimulated with white flashes of 4 ms duration, ranging in intensity from −5 to −2 log cd·s/m^2^ and ERGs were recorded as the average response to multiple consecutive flashes at each intensity (100 flashes at −5 to −4 log cd·s/m^2^, 30 flashes at −2 log cd·s/m^2^). Recordings included a 100 ms pre-stimulus baseline with data collected up to 500 ms after stimuli onset. Results from each eye were considered independently. ERG data from the Espion software V6.68.2 (Diagnosis) were exported into Microsoft Excel (Microsoft, Redmond, WA, USA) for analysis. Positive scotopic threshold response (pSTR) was determined by the peak amplitude of the waveform to baseline. The a-wave amplitude was determined by the peak amplitude of the trough to baseline. The b-wave amplitude was calculated from the a-wave trough to the b-wave peak. Latencies were determined as the time intervals from stimulus onset to peak/trough.

### 4.6. Optic Nerve Histology and Analysis

Mice were euthanized by carbon dioxide inhalation, then cardiac perfused with 10 mL of phosphate-buffered saline (PBS) followed by 20 mL of 4% paraformaldehyde (PFA) in PBS. Optic nerves (ONs) were cut at the optic chiasm and proximal to the globe, post-fixed in 4% PFA and 1% glutaraldehyde in PBS, and stored for 1–3 months. ONs were post-fixed in 2% osmium and embedded in epoxy resin (Embed 812/Araldite resin, Electron Microscopy Sciences) as previously described [13]. The length of the optic nerve stump remaining attached to the eye was measured using a digital caliper (Instant Read-Out Digital Calipers, Electron Microscopy Sciences, Hatfield, PA, USA). To normalize the distance from the eye at which the optic nerve sections were taken, ON blocks were trimmed so that the stump length plus the cut in distance along the optic nerve was 1.5 mm from the globe. Using an ultra-microtome (Leica EM UC7, Wetzlar, Germany), 1 μm thick cross sections were cut and stained with paraphenylenediamine (PPD), which darkly stains axoplasm of degenerating axons [48], and mounted with Permount Mounting Medium (Thermo Fisher Scientific, Waltham, MA, USA). Mosaic images of stained optic nerve cross sections were acquired with an oil immersion 100X/1.45NA objective with an SLR DS-Ri2 camera on a Nikon upright microscope using NIS-elements software 6.10.01 (Nikon, Tokyo, Japan). ON area was determined by drawing a polygon around the nerve, excluding the pia mater, using Fiji (version 2.9.0) [49]. ON axons were manually counted [50] and analyzed using AxoNet2.0 [51].

### 4.7. Statistical Analysis

Experiments were conducted in a masked fashion. Statistical analyses including Welch’s two-tailed *t*-test and linear regression were performed using GraphPad Prism version 9.0 (GraphPad Software, San Diego, CA, USA). Data are presented as mean ± standard deviation (SD). *p*-values and the number of data points (n) are indicated in the figures and figure legends.

## Figures and Tables

**Figure 1 ijms-26-09221-f001:**
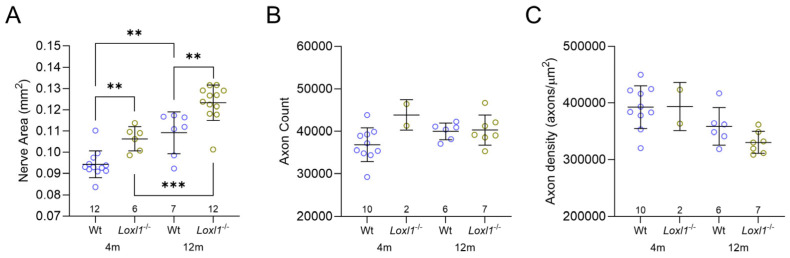
*Loxl1*^−/−^ mice have enlarged optic nerves compared with wt mice. (**A**) At 12 months old, *Loxl1*^−/−^ mice had enlarged optic nerve cross-sectional areas compared with wt mice at both 4 and 12 months of age (4 m: 106 ± 5.7 µm^2^ (*Loxl1*^−/−^) vs. 94.4 ± 6.3 µm^2^ (wt); 12m: 123 ± 8.4 µm^2^ (*Loxl1*^−/−^) vs. 109 ± 9.9 µm^2^ (wt); ** *p* < 0.01, *** *p* < 0.001). (**B**) No decrease in total axon count was seen in *Loxl1*^−/−^ mice. (**C**) There was a borderline decrease in axon density in *Loxl1*^−/−^ mice at 12 months old (*p* = 0.07). Genotypes and ages of mice are denoted under the horizontal axis, and number of eyes in each grouping are indicated above the horizontal axis.

**Figure 2 ijms-26-09221-f002:**
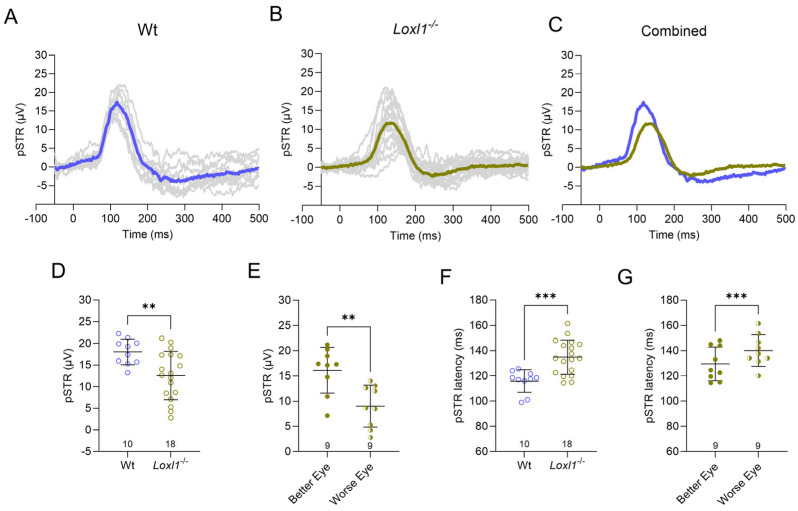
*Loxl1*^−/−^ mice have reduced RGC function compared with wt mice at 12 months of age. Positive scotopic threshold response (pSTR) ERG waveforms of individual mice (gray traces in (**A**,**B**)) and average waveforms (blue trace in (**A**) and green trace in (**B**)) in response to flash intensities of −5 log cd·s/m^2^ are shown. The decreased amplitude and increased latency of pSTR in *Loxl1*^−/−^ mice (green) compared with wt mice (blue) is demonstrated in (**C**). The statistically significant reduction in pSTR amplitude and increase in latency in *Loxl1*^−/−^ compared with wt mice are quantified in (**D**,**F**). *Loxl1*^−/−^ eye asymmetry is shown in (**E**,**G**). Genotype is indicated at the top of (**A**,**B**), and the number in each group is indicated above the horizontal axis in (**D**,**E**). ** *p* < 0.01, *** *p* < 0.001.

**Figure 3 ijms-26-09221-f003:**
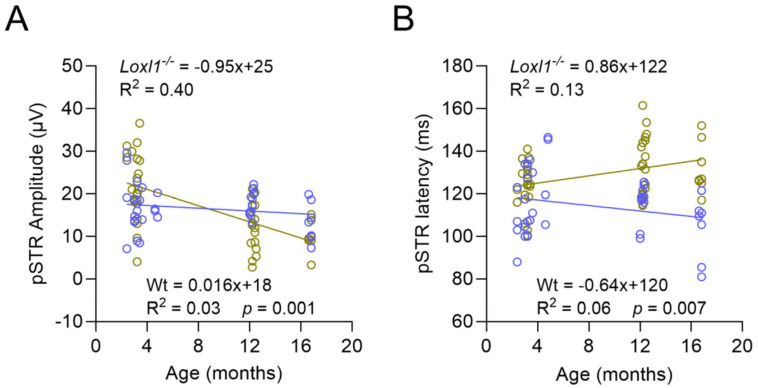
*Loxl1*^−/−^ mice have age-related progressive reduction in RGC function compared with wt mice. The average amplitude (**A**) and latency (**B**) of pSTR at 4-, 12-, and 16-month-old wt (blue in (**A**,**B**)) and *Loxl1*^−/−^ (green in (**A**,**B**)) mice. Linear regressions (equations for *Loxl1*^−/−^ on the top and wt on the bottom of the graph in (**A**,**B**)) reveal progressive reductions in pSTR amplitude (**A**) and increased latency (**B**) in *Loxl1*^−/−^ mice, whereas wt mice remain relatively stable. The magnitudes of age-related change in both pSTR amplitude and latency are significantly different between *Loxl1*^−/−^ and wt mice (*p* = 0.001 for pSTR amplitude in A and *p* = 0.007 for pSTR latency in (**B**)).

**Figure 4 ijms-26-09221-f004:**
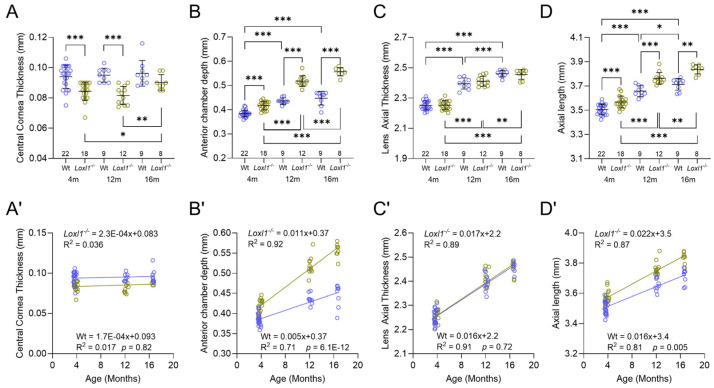
Age-related biometric changes in *Loxl1*^−/−^ and wt mice. Differences between wt (blue) and *Loxl1*^−/−^ mice (green) in CCT (**A**), ACD (**B**), lens axial thickness (LAT) (**C**), and AL (**D**) are shown in mice at 4, 12, and 16 months of age. Most measurements between the 3 ages are significantly different (* *p* < 0.05, ** *p* < 0.01, *** *p* < 0.001). Linear regressions (equations for *Loxl1*^−/−^ on the top and wt on the bottom of the graph in (**A’**–**D’**)) reveal relatively stable CCT (**A’**) in both wt and *Loxl1*^−/−^ mice, whereas progressive deepening of ACD (**B’**), thickening of LAT (**C’**), and elongation of AL (**D’**) occurs in both groups. The magnitudes of age-related change in both ACD and AL are significantly different between *Loxl1*^−/−^ and wt mice (*p* = 6.1 × 10^−12^ for ACD in (**B’**) and *p* = 0.005 for AL in (**D’**)).

**Figure 5 ijms-26-09221-f005:**
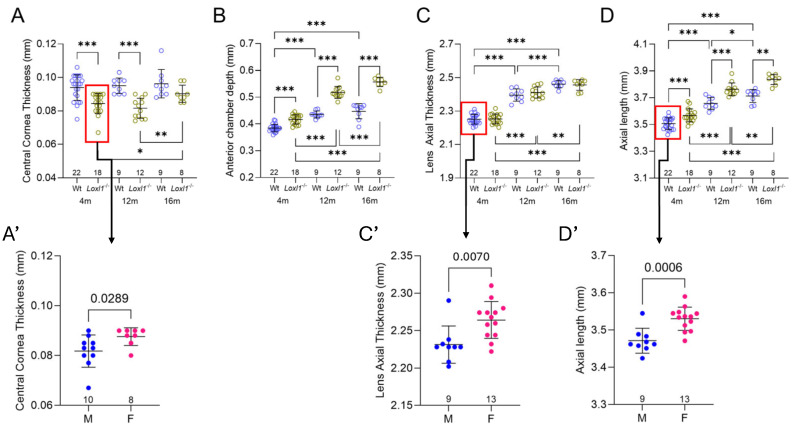
Biometric measurement differences related to sex. The differences between male (blue) and female (pink) mice are statistically significant for CCT (*p* = 0.0289 in (**A’**)) in *Loxl1*^−/−^ mice (green circles shown in (**A’**)), LT (*p* = 0.007 in (**C’**)) in wt mice (blue circles shown in (**C’**)), and AL (*p* = 0.0006 in (**D’**)) in wt mice (blue circles shown in (**D’**)). (**A**), (**B**), (**C**) and (**D**) are identical to (**A**), (**B**), (**C**) and (**D**) in Figure 4, respectively. * *p* < 0.05, ** *p* < 0.01, *** *p* < 0.001.

**Figure 6 ijms-26-09221-f006:**
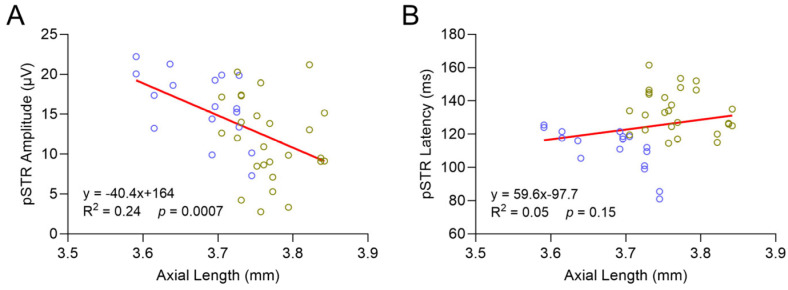
Increased axial length (AL) elongation was correlated with decreased pSTR amplitude. Analysis of combined 12- and 16-month-data from both genotypes (wt: blue; *Loxl1*^−/−^: green) shows significant correlation of decreased pSTR amplitude with increased AL ((**A**); *p* = 0.0007), but no significant correlation between pSTR latency with globe elongation ((**B**); *p* = 0.15).

**Figure 7 ijms-26-09221-f007:**
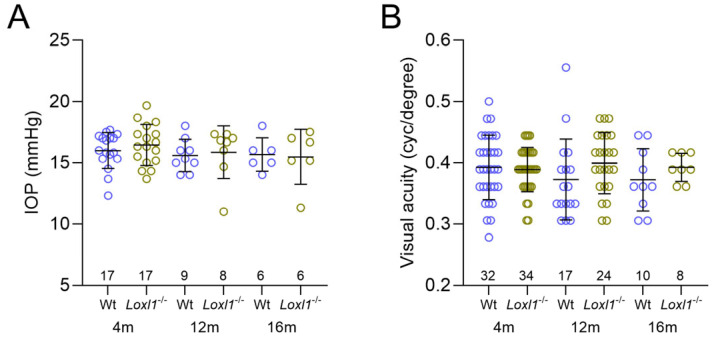
No change over time or between genotypes in IOP or visual acuity. IOP of wt (blue) and *Loxl1*^−/−^ (green) mice is similar at all 3 ages and remains relatively constant (**A**). Visual acuity, measured by optomotor reflex, remains similar between groups at all 3 ages and relatively stable throughout (**B**). Genotypes and ages of mice are denoted under the horizontal axes and the number of eyes in each genotype and age group are indicated above the horizontal axes.

## Data Availability

The raw data supporting the conclusions of this article will be made available by the authors on request.

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
