# Peer review of "Progressive Ocular Axial Elongation and Retinal Ganglion Cell Degeneration in Mice with Elastic Fiber Disorder"

_ijms, 2025, doi:10.3390/ijms26189221_

Round 1

Reviewer 1 Report

Comments and Suggestions for Authors

Title: Progressive Ocular Axial Elongation and Retinal Ganglion Cell Degeneration in Mice with Elastic Fiber Disorder

This study investigates the relationship between axial elongation and glaucoma using the Lox1 mouse model. The experimental design is appropriate and the methodology is generally well executed. However, there are some concerns regarding the primary factor and its interpretation.

Major Comments

  1. Since LOX1 is a gene strongly associated with PXG, the main exposure factor here is the Lox1 mutation itself. Therefore, the conclusion that axial elongation is the primary cause of glaucoma may be open to alternative interpretations. If the degeneration of ganglion cells is related to growth (myopia), it is difficult to reconcile the differences in axial length between cases and controls at 4 months of age.

  2. A schematic figure explaining the experimental design would be helpful for clarity.

  3. To support the claim of an elastic fiber disorder, it is necessary to provide histological evidence such as scleral tissue or connective tissue staining.

Minor Comments

  1. The number of mice used in the experiments should be clearly described in the Methods section.

Reviewer 2 Report

Comments and Suggestions for Authors

The manuscript is interesting. This experimental model can be used for studying several ocular diseases including glaucoma, and pathologic myopia. However, several issues should be well-addressed.

1) Male's and female's outcome differences?

2) Light condition during development should be clearly stated.

3) Images are all missing It is important to see the data with quantification. 

4) Depending on the retinal layers, the damage level should be presented.

5) Any biomolecular data? Even Loxl1 knock out data are missing. Furthermore, it is good to include any biomolecular data with aging.

6) This knockout system has any behavioral change?

7) Body weight and systemic metabolic changes are recommended to be added.

Round 2

Reviewer 1 Report

Comments and Suggestions for Authors

The revision has clarified the content. Some typos are necessary for correction (ex.  Figure 65. Figure 76.)

Comments on the Quality of English Language

Some expressions would benefit from revision (line 116-118) "Glaucoma is an age-related neurodegenerative disease characterized by loss of RGCs[18]." is not fitted to result section. 

Author Response

The revision has clarified the content. Some typos are necessary for correction (ex.  Figure 65. Figure 76.)

We sincerely thank the reviewer for such careful read and have revised the figure legends as attached here. Please note the new figure legends should be: Figure 65. Increased axial length (AL) elongation was correlated with decreased pSTR amplitude. (line 186); Figure 76. No change over time or between genotypes in IOP or visual acuity. (line 198)

Some expressions would benefit from revision (line 116-118) "Glaucoma is an age-related neurodegenerative disease characterized by loss of RGCs[18]." is not fitted to result section. 

We thank the reviewer for this thoughtful comment. We agree that such broad statement on glaucoma seems more appropriate if placed in Introduction and/or Discussion. However, we believe it is necessary in the context of what we report in this specific section of Results. It sets the stage for what we intend to report on the age-related phenotype of Loxl1-/- mice shown in the following sentence: "We next examined if the reduced RGC function in Loxl1-/- mice is age related." (line 118). We respectfully request the editor grant us the permission to keep this sentence.

Reviewer 2 Report

Comments and Suggestions for Authors

The attached response letter has been read and it seems the revised manuscript addressed my comment. But, I can't find the figures in the manuscript. It seems that it is my own technical problem (?), as this manuscript has been quality-passed? Anyway, I can't find figures in the manuscript. But, the raised concerns seem to be addressed based on the response letter.

Author Response

The attached response letter has been read and it seems the revised manuscript addressed my comment. But, I can't find the figures in the manuscript. It seems that it is my own technical problem (?), as this manuscript has been quality-passed? Anyway, I can't find figures in the manuscript. But, the raised concerns seem to be addressed based on the response letter.

We are sorry for such frustrating confusion. We checked again and noticed that the revised figures are embedded in the manuscript. We attached revised figures (new figure 5 and new supplemental figure) here in PDF with the hope that it will be viewable to the reviewer. 
